# Nicotine Exposure during Adolescence Leads to Changes of Synaptic Plasticity and Intrinsic Excitability of Mice Insular Pyramidal Cells at Later Life

**DOI:** 10.3390/ijms23010034

**Published:** 2021-12-21

**Authors:** Hiroki Toyoda, Kohei Koga

**Affiliations:** 1Department of Oral Physiology, Osaka University Graduate School of Dentistry, Suita 565-0871, Japan; 2Department of Neurophysiology, Hyogo College of Medicine, Nishinomiya 663-8501, Japan; kkoga@hotmail.co.jp

**Keywords:** nicotine, insular cortex, pyramidal cell, synaptic plasticity, intrinsic excitability

## Abstract

To find satisfactory treatment for nicotine addiction, synaptic and cellular mechanisms should be investigated comprehensively. Synaptic transmission, plasticity and intrinsic excitability in various brain regions are known to be altered by acute nicotine exposure. However, it has not been addressed whether and how nicotine exposure during adolescence alters these synaptic events and intrinsic excitability in the insular cortex in adulthood. To address this question, we performed whole-cell patch-clamp recordings to examine the effects of adolescent nicotine exposure on synaptic transmission, plasticity and intrinsic excitability in layer V pyramidal neurons (PNs) of the mice insular cortex five weeks after the treatment. We found that excitatory synaptic transmission and potentiation were enhanced in these neurons. Following adolescent nicotine exposure, insular layer V PNs displayed enhanced intrinsic excitability, which was reflected in changes in relationship between current strength and spike number, inter-spike interval, spike current threshold and refractory period. In addition, spike-timing precision evaluated by standard deviation of spike timing was decreased following nicotine exposure. Our data indicate that adolescent nicotine exposure enhances synaptic transmission, plasticity and intrinsic excitability in layer V PNs of the mice insular cortex at later life, which might contribute to severe nicotine dependence in adulthood.

## 1. Introduction

Effective treatment of nicotine dependence is crucial for decreasing the severe morbidity and mortality in relation with tobacco smoking. Although many efforts have been made to elucidate the synaptic and cellular mechanisms for nicotine addiction in rodents, there are currently still few effective treatments [1]. To produce satisfactory treatment strategies of nicotine addiction, the synaptic and cellular mechanisms should be further investigated, and treatment strategy of nicotine addiction on the synaptic and cellular level may be more potential options in the future.

Adolescence is a sensitive period for drug consumption and addiction both in animals and humans [2,3]. At this stage in life, a critical phase of the neurodevelopmental process, abuse of drugs including tobacco smoking can induce brain plasticity such as long-lasting changes in neural circuitry both in animals and humans [4,5,6]. Chronic smoking during adolescence in humans is detrimental to cognitive function and contributes to the occurrence of cognitive deficits [7,8]. Adolescent nicotine exposure in rats caused a long-lasting impairment in attention and elevated impulsive behavior 5 weeks after the treatment [9]. At the neuron level, adolescent nicotine exposure in rats was reported to increase spike-timing dependent plasticity in layer V PNs of the prefrontal cortex in later life [10]. Currently, little study has been conducted regarding the effects of adolescent nicotine exposure on synaptic and cellular functions at later life in other rodent brain regions associated with nicotine addiction.

In recent years, there is an enhanced understanding of the critical role of the insular cortex in addiction to cigarette smoking [11,12,13]. In previous human studies, smokers with brain damage to the insula experienced reduced desire to smoke compared with those with brain damage not involving the insula [14]. It has also been demonstrated that damage to the insula in humans was correlated with increased odds of smoking cessation and complete abstinence from nicotine-related products [15]. In addition, several animal studies have shown that nicotine-taking and nicotine-seeking behaviors were decreased when the rat granular and agranular insular cortices were inactivated [16,17,18]. Our study has previously shown that in layer V PCs of the adolescent mice insular cortex, acute nicotine application increased both glutamatergic and GABAergic transmission through activation of β2-containing nicotinic acetylcholine receptors (nAChRs), whereas it suppressed LTP through increasing GABAergic transmission via action of β2-containing nAChRs expressed in non fast-spiking neurons [19]. However, it remains unknown how nicotine exposure during adolescence modulates synaptic transmission and plasticity in insular layer V PNs at later life.

The excitability of neurons depends on not only synaptic inputs but also intrinsic excitability. A previous study has demonstrated that in vivo nicotine administration for 1 week in young adult rats increased the intrinsic excitability of hippocampal CA1 neurons [20]. The effect of chronic nicotine treatment was long-lasting and the enhanced excitability by nicotine may be a cellular mechanism for nicotine dependence [20]. However, little is known about how adolescent nicotine exposure alters neuronal intrinsic excitability at later life in rodent brain areas involved in nicotine addiction.

Here, we first studied effects of adolescent nicotine exposure on synaptic transmission and plasticity in layer V PNs of the mice insular cortex 5 weeks after the treatment. We also studied the intrinsic excitability by assessing relationship between current strength and spike number, inter-spike interval, spike current threshold and refractory period. Furthermore, the spike timing precision was investigated by calculating standard deviation of spiking timing (SDST). Our data indicate that synaptic events and intrinsic excitability of layer V PNs in the mice insular cortex were enhanced following nicotine exposure during adolescence, which might contribute to severe nicotine dependence at later life.

## 2. Results

### 2.1. Adolescent Nicotine Exposure Increases Excitatory Synaptic Transmision and Potentiation

First, we examined whether and how nicotine exposure during adolescence alters excitatory synaptic transmission and plasticity in layer V PNs of the mouse insular cortex 5 weeks after the treatment. Whole-cell patch-clamp recordings were made on visually identified PNs in layer V of the insular cortex. These neurons were further identified by examining the firing pattern in response to depolarizing current injections. The common firing patterns of PNs displayed spike frequency adaptation [19]. The frequency and amplitude of spontaneous excitatory postsynaptic currents (sEPSCs) obtained from layer V PNs of nicotine-treated mice (frequency of 5.5 ± 2.3 Hz, amplitude of 18.1 ± 0.6 pA; n = 16 neurons/7 mice) were significantly larger (frequency and amplitude: unpaired *t*-test, *t* (31) = −3.38, *p* = 0.002 and unpaired *t*-test, *t* (31) = −4.03, *p* < 0.001) than those obtained from saline-treated mice (frequency of 3.5 ± 0.3 Hz, amplitude of 15.4 ± 0.4 pA; n = 17 neurons/6 mice) (Figure 1A–C). These data suggest that adolescent nicotine exposure increased glutamatergic transmission in layer V PNs of the insular cortex at later life.

We next examined whether and how nicotine exposure during adolescence alters synaptic potentiation using saline-treated and nicotine-treated mice 5 weeks after the treatment. After identification of PNs, control current responses were recorded for 10 min. Then, we induced LTP by pairing presynaptic stimulation (80 pulses at 2 Hz) with postsynaptic depolarization (+30 mV) [19,21]. The LTP-inducing stimulation caused a significant long-lasting potentiation of synaptic responses in saline-treated mice (129.5 ± 5.6% of baseline, n = 9 neurons/6 mice; paired *t*-test: *t* (8) = −3.91, *p* = 0.004) and nicotine-treated mice (155.8 ± 5.8% of baseline, n = 8 neurons/7 mice; paired *t*-test: *t* (7) = −7.27, *p* < 0.001) (Figure 2A–D). The increment of synaptic potentiation was significantly larger in nicotine-treated mice than in saline-treated mice (unpaired *t*-test: *t* (15) = −2.62, *p* = 0.019) (Figure 2D). These data suggest that synaptic potentiation is enhanced in layer V PNs of the mouse insular cortex 5 weeks after the adolescent nicotine exposure.

### 2.2. Adolescent Nicotine Exposure Enhancecs Spiking Ability

We next investigated whether and how nicotine exposure during adolescence alters spiking ability in layer V PNs of the insular cortex 5 weeks after the treatment. To examine properties of spike firings, voltage responses to depolarizing current pulses were measured from layer V PNs in saline-treated and nicotine-treated mice under the current-clamp condition (Figure 3A). The relationship between current strength and spike number revealed that the number of spikes was significantly greater in nicotine-treated mice (n = 13 neurons/7 mice) than in saline-treated mice (n = 13 neurons/6 mice) at current intensities of 60 to 240 pA (Two-way repeated measures ANOVA followed by Fisher’s protected least significant difference (LSD) post-hoc test, F(7,210) = 5.94: 30 pA, *p* = 0.419; 60 pA, *p* = 0.013; 90 pA, *p* < 0.001, 120 pA, *p* < 0.001, 150 pA, *p* < 0.001, 180 pA, *p* < 0.001, 210 pA, *p* < 0.001, 240 pA, *p* < 0.001) (Figure 3B).

Furthermore, we tested the effects of adolescent nicotine exposure on inter-spike interval. The inter-spike interval is the time between two consecutive spikes in a spike train. The first inter-spike interval that is the interval between the first and second spikes in PNs obtained from nicotine-treated mice (31.7 ± 2.8 ms, n = 13 neurons/7 mice) was not significantly different (unpaired *t*-test: *t* (24) = 0.40, *p* = 0.696) from those obtained from saline-treated mice (33.2 ± 2.6 ms, n = 13 neurons/6 mice) (Figure 3C,D). However, the steady-state inter-spike interval which was defined as the mean inter-spike interval in the steady-state firings during the last 500 ms of the current pulse was significantly shorter (unpaired *t*-test: *t* (24) = 3.88, *p* < 0.001) in PNs obtained from nicotine-treated mice (65.5 ± 3.3 ms, n = 13 neurons/7 mice) compared with those obtained from saline-treated mice (85.7 ± 4.0 ms, n = 13 neurons/6 mice) (Figure 3C,D). In addition, the spike frequency ratio which was obtained by dividing the steady-state inter-spike interval with the first inter-spike interval was calculated [22]. The spike frequency ratio was significantly smaller (unpaired *t*-test: *t* (24) = 2.26, *p* = 0.034) in PNs of the nicotine-treated mice (2.1 ± 0.2, n = 13 neurons/7 mice) than in those of the saline-treated mice (2.9 ± 0.3, n = 13 neurons/6 mice) (Figure 3E). These findings indicate that nicotine exposure during adolescence increases spiking ability in layer V PNs of the insular cortex 5 weeks after the treatment.

### 2.3. Adolescent Nicotine Exposure Changes Spike Current Threshold and Input Resistance

In order to explain the enhanced spiking ability by adolescent nicotine exposure, we studied the membrane properties of layer V PNs of the insular cortex. A spike generation is induced when summation of synaptic inputs onto postsynaptic cells reaches spike threshold. Thus, the spike threshold reflects how easily neurons turn synaptic inputs into spikes. In the present study, the spike threshold was accessed using current threshold. The spike current threshold was determined by increasing current stimulation intensity until evoking a spike with a probability of 50%. The spike current threshold was significantly decreased (unpaired *t*-test: *t* (28) = 2.73, *p* = 0.011) in PNs obtained from nicotine-treated mice (488 ± 37 pA, n = 15 neurons/7 mice) compared to those obtained from saline-treated mice (712 ± 70 pA, n = 15 neurons/6 mice) (Figure 4A,B). No significant difference was found in resting membrane potential (unpaired *t*-test: *t* (28) = −1.93, *p* = 0.064) between saline-treated mice (−70.4 ± 1.1 mV, n = 15 neurons/6 mice) and nicotine-treated mice (–67.4 ± 1.2 mV, n = 15 neurons/7 mice) (Figure 4C). In contrast, the input resistance was significantly larger (unpaired *t*-test: *t* (28) = 2.63, *p* = 0.013) in nicotine-treated mice (205 ± 24 MΩ, n = 15 neurons/7 mice) compared to saline-treated mice (137 ± 9 MΩ, n = 15 neurons/6 mice) (Figure 4D). These data suggest that the decreased spike current threshold and increased input resistance contribute to the enhanced spiking ability in layer V PNs of nicotine-treated mice.

### 2.4. Adolescent Nicotine Exposure Decreases Refractory Period

The refractory period which is associated with maximal firing rate of neurons is the shortest interval between two action potentials at a given strength of paired current pulses [23]. The refractory period was obtained by measuring the time from a complete spike to its next one at 50% probability when the interval of paired current pulses were changed [24]. The refractory period was significantly shorter (unpaired *t*-test: *t* (28) = 3.50, *p* = 0.002) in PNs obtained from nicotine-treated mice (6.7 ± 0.6 ms, n = 16 neurons/7 mice) compared to those obtained from saline-treated mice (8.2 ± 0.4 ms, n = 14 neurons/6 mice) (Figure 5A,B). This finding suggests that the adolescent nicotine exposure decreases refractory period, which is associated with the enhanced intrinsic excitability in layer V PNs of the insular cortex.

### 2.5. Adolescent Nicotine Exposure Decreases Precision of Spike Timing

In addition to the spiking ability, spike timing plays crucial roles in neuronal coding and information processing [25]. Spike timing precision contributes to many physiological and pathological conditions [26,27]. To examine whether and how adolescent nicotine exposure affects precision of spike timing, standard deviation of spike timing (SDST) was calculated by using 20 recordings of spike trains evoked by repetitive stimuli. The first to tenth SDST were compared between PNs in saline-treated mice and those in nicotine-treated mice. As shown in the relationship between spike number and SD of spike timing (Figure 6B), SDSTs of spike timing from fifth to tenth spikes were significantly greater in nicotine-treated mice (n = 15 neurons/7 mice) than in saline-treated mice (n = 16 neurons/6 mice) (Two-way repeated measures ANOVA followed by post-hoc LSD test, F(9,243) = 7.40: 1st, *p* = 0.922; 2nd, *p* = 0.855; 3rd, *p* = 0.415; 4th, *p* = 0.115; 5th, *p* = 0.040; 6th, *p* = 0.005; 7th, *p* < 0.001; 8th, *p* < 0.001; 9th, *p* < 0.001; 10th, *p* < 0.001). These observations indicate that the precision of spike timing was decreased in layer V PNs of nicotine-treated mice. The decreases in spike timing precision might be involved in pathological condition caused by nicotine exposure.

## 3. Discussion

The present study demonstrated that nicotine exposure during adolescence enhanced synaptic transmission, plasticity and intrinsic excitability in layer V PNs of the mice insular cortex at later life. In addition, the precision of spike timing in these neurons was decreased following adolescent nicotine exposure. Our findings may be helpful for understanding the mechanisms which might contribute to severe nicotine dependence in adulthood.

Neurons transmit information through complex patterns of spikes which are mediated by spatial and temporal summation of synaptic inputs. Synaptic transmission represents an essential component in neuronal information processing [28]. Substantial studies have been performed focusing on the effects of nicotine on synaptic transmission in the rodent cortex. In layer V PNs of the rodent prefrontal and insular cortices, excitatory synaptic transmission was markedly enhanced by acute bath application of nicotine [19,29,30]. Similar to these results, adolescent nicotine exposure increased excitatory synaptic inputs onto layer V PNs of the mice insular cortex 5 weeks after the treatment. The increased excitatory synaptic inputs onto these neurons definitely cause more spikes, and thus lead to neuronal hyperexcitability. Furthermore, we found that adolescent nicotine exposure increased synaptic potentiation in layer V PNs of the mice insular cortex 5 weeks after the treatment. Consistent with this finding, in layer V PNs of the rat prefrontal cortex, synaptic potentiation was shown to be enhanced 5 weeks after nicotine exposure during adolescence [10]. Besides, unlike with nicotine treatment during adolescence, nicotine exposure during adulthood was demonstrated not to lead to lasting changes in spike-timing dependent plasticity in the rat prefrontal layer V PNs 5 weeks after the treatment [10]. These findings strongly suggest that adolescence is a critical period of vulnerability for long-term effects of nicotine on the cortex in rodents. The repeated exposure of nicotine during adolescence may affect the maturation of mice insular circuitry and lead to changes in the insular circuitry that may last for a lifetime. It has been shown that the enhanced spike-timing dependent plasticity caused by adolescent nicotine exposure in the rat prefrontal layer V PNs was mediated by a long-term decrease in the expression of synaptic metabotropic glutamate receptor 2 (mGluR2) [31], which was present in presynaptic terminal and reduces excitatory synaptic transmission [32,33]. Thus, it is possible that adolescent nicotine exposure reduces synaptic mGluR2 expression in the mice insular cortex at later life, which may increase synaptic potentiation in insular layer V PNs. Another possible mechanism for the facilitation of synaptic potentiation following adolescent nicotine exposure could involve an elevated expression of nAChRs. In layer V PNs of the mice insular cortex, functional nAChRs are present [19], and it was demonstrated that both α7- and β2-containing nAChRs were upregulated by chronic nicotine exposure on the rodent cortex [34,35]. Thus, adolescent nicotine exposure might have increased expression of functional nAChRs in mice insular layer V PNs, which may lead to enhanced synaptic potentiation.

In the present study, we have shown that neuronal spiking ability was increased after adolescent nicotine exposure as revealed by the results obtained from the measurements of the relationship between current strength and spike number and inter-spike intervals. The relationship between current strength and spike number reflects neuronal responses, and it has been shown that the modulation of its relationship occurred when the motor cortex displayed plasticity [36]. Several factors are known to modulate the relationship between current strength and spike number in cortical neurons. One of main factors is synaptic input fluctuations. Cortical neurons spontaneously discharge in vivo, causing irregular background synaptic noise to other neurons [37]. This induces synaptic input fluctuations in the membrane potential at the postsynaptic membrane and affects the relationship between current strength and spike number [38]. Because application of nicotine was demonstrated to increase synaptic noise in the rodent prefrontal cortex and ventral tegmental area [30,39], enhancement of synaptic input fluctuations by nicotine is likely to contribute to the enhanced spiking ability. As another factor, release of neurotransmitter is known to modulate the relationship between current strength and spike number. For instance, it has been demonstrated that serotonin (5-HT) markedly enhanced the slope of the firing rate-current curve through activation of 5-HT type 2 (5-HT_2_) receptors in layer V PNs of the rat prefrontal cortex [40]. The modulation by 5-HT was attributed to reduction of afterhyperpolarization and induction of slow afterdepolarization, but not to changes in membrane potential, input resistance or action potential [40]. Also, it has been shown that dopamine increased neuronal excitability by shifting the frequency-current curve to the left through activation of D1 receptors in layer V PNs of the rat prefrontal cortex [41]. The modulation by dopamine was attributed to reduction of slow afterhyperpolarization [41].

We also evaluated the intrinsic excitability by the measurements of spike current threshold and refractory period. Following nicotine exposure during adolescence, the spike current threshold and refractory period in layer V PNs were significantly decreased compared to those obtained from saline-treated mice. The spike current threshold is a critical factor in determining the relationship between current strength and spike number, and changes of spike current threshold can modulate the frequency–current curve [42]. The refractory period reflects time latency to elicit another spike caused by stimulus, and is also a pivotal element for spiking ability [43]. Thus, the decreases in the spike current threshold and refractory period are likely to be involved in the leftward shift of the relationship between current strength and spike number, and these changes reflect well the enhanced spiking ability. A previous study has shown that the enhanced excitability of rat CA1 pyramidal neurons following nicotine treatment was due to a decrease in Ba^2+^-sensitive K^+^ currents [20]. Thus, the increased input resistance in mice layer V PNs following nicotine exposure observed in this study is possibly to be mediated by a decrease in Ba^2+^-sensitive K^+^ currents, and this may underlie the enhanced excitability of these neurons. However, it would be necessary to elucidate the mechanism how adolescent nicotine exposure enhances intrinsic excitability in mice insular layer V PNs at later life. Figure 7 shows a possible model for long-term effects of adolescent nicotine exposure in layer V PNs of the mice insular cortex.

There is growing evidence that spike timing plays an essential role in neuronal information processing [25,44]. Neurons can change spike timing depending on the sensory inputs, and spike timing is a significant element in temporal coding [45]. In the cortex which receives multiple synaptic inputs, the precision of spike timing is a key factor for accurate encoding of sensory information, especially for the taste, auditory and visual systems [46,47,48]. Previous studies have demonstrated that the reduced spike timing precision was associated with some pathological conditions such as epilepsy and chronic pain [24,26,49]. In the rat epileptic hippocampus, a reduction of spike timing was correlated with the decreased spatial synchronization and the occurrence of fast ripples [26]. In neurons of the mice anterior cingulate cortex, peripheral inflammation by complete Freund’s adjuvant decreased the precision of spike timing [49]. Moreover, neuropathic pain caused by spared nerve injury decreased the precision of spike timing in pyramidal neurons of the mice anterior cingulate cortex [24]. Consistent with these findings, we also found in the present study that adolescent nicotine exposure decreased the precision of spike timing in layer V PNs of the mice insular cortex 5 weeks after the treatment. Although it remains largely unknown how the reduced spike timing precision is involved in the pathological conditions including chronic pain and nicotine addiction, this might be a critical issue for elucidating cellular mechanisms of nicotine addiction.

There are crucial issues associated with this study which are required for discussion. First, the nicotine dose used in this study was relatively high, compared to those in previous studies using C57BL/6 mice [50,51]. It is generally known that mice are less sensitive to acute effects of nicotine than rats. For instance, the higher brain levels of nicotine are necessary to achieve comparable antinociceptive effects in mice compared to rats [52]. In addition, the plasma half-life of nicotine in mice (6–7 min) is shorter than those in rats (45 min) and humans (2 h) [53]. Based on these findings, a relatively higher concentration of nicotine was used in the present study. However, the steady-state level of nicotine in mice may not have been necessarily comparable to that of human smokers. Thus, it would be necessary to test lower concentrations of nicotine for injection in the future studies. Second, it is not clear whether or not the present animal findings can apply to human findings. However, it has been accepted that adolescence is a sensitive period for drug addiction both in animals and humans [1,2]. Although it is difficult in humans to confirm similar behavioral effects of teen nicotine exposure, observational studies indicate similarities between animals and humans [54]. Therefore, our findings in mice could provide important clues for understanding mechanisms and therapeutics in humans. However, the present results obtained from mice should be interpreted with caution. This is because nicotine delivery via subcutaneous injection and duration of nicotine exposure (7 days) do not truly reflect the common pattern of nicotine delivery in humans. In addition, there is not always concordance between animal and human findings in the central nervous system. For example, brain imaging of α4β2 nAChRs using specific probes revealed that brain distribution of α4β2 nAChRs varied between mice, rats, monkeys and humans [55]. Therefore, it may be difficult to directly apply the present results to humans.

In summary, our findings suggest that adolescent nicotine exposure causes enhanced synaptic transmission, plasticity and intrinsic excitability in layer V PNs of the mice insular cortex at later life. These changes in synaptic events and intrinsic membrane properties may help to assist for understanding mechanisms of severe nicotine dependence caused by adolescent nicotine exposure.

## 4. Materials and Methods

All experiments were performed according to the National Institutes of Health Guide for the Care and Use of Animals. All animal experiments and the protocol were approved by the Institutional Animal Care and Use Committee at Osaka University Graduate School of Dentistry (DENT31-003-1). All efforts were made to decrease the number of animals used and animal suffering.

### 4.1. Animals

Five-week-old C57BL/6J male mice (Japan SLC, Hamamatsu, Japan) were used in the present study to rule out the possible sex differences in effects of nicotine in animals [56]. The mice were housed three per cage and maintained at temperature of 23 ± 2 °C under 12 h light/dark cycle. The mice can freely access to food and water. Effects of nicotine on synaptic transmission, plasticity and intrinsic excitability were examined using nicotine hydrogen tartrate salt as this form of nicotine is considered to be more stable than nicotine free base. Either nicotine (9.0 mg/kg/day of nicotine hydrogen tartrate salt: 3.16 mg/kg/day of nicotine base) or saline was subcutaneously injected two times per day (at 8:00 and 20:00) for 7 days. This concentration of nicotine has been shown to correlate with the rat plasma nicotine concentration, similar to that reported in human smokers [57]. After the 7 days treatment, the injections stopped, and the electrophysiological experiments were conducted 5 weeks after the treatment. Before and after the nicotine treatment, body weight was measured weekly. The nicotine-treated mice (n = 7) had significantly less body weight than the saline-treated mice (n = 6), just after the treatment. However, 1 to 5 weeks after the nicotine treatment, there were no significant differences in body weight between two groups (Appendix A).

### 4.2. Slice Preparation

Saline-treated and nicotine-treated mice were used for this study. Five weeks after saline or nicotine exposure, coronal brain slices containing the agranular insular cortex were prepared as described in our study [19]. They were decapitated under deep anesthesia with isoflurane. The brain was carefully removed from the skull and immediately placed into ice-cold solution (in mM): 210 sucrose, 2.5 KCl, 2.5 MgSO_4_, 1.25 NaH_2_PO_4_, 26 NaHCO_3_, 0.5 CaCl_2_ and 50 D-glucose (pH 7.3). Coronal brain slices at the thickness of 300 μm were obtained using a vibratome (Linearslicer Pro 7, Dosaka EM, Kyoto, Japan). Slices were transferred to a submerged holding chamber containing the cutting solution at 32 °C for 30 min. After that, slices were transferred to a submerged holding chamber containing artificial cerebrospinal fluid (aCSF) at 32 °C for 30 min. aCSF contained following (mM): 126 NaCl, 3 KCl, 1 MgSO_4_, 1.25 NaH_2_PO_4_, 26 NaHCO_3_, 2 CaCl_2_ and 10 D-glucose. Then, the slices were transferred to a submerged holding chamber containing aCSF and maintained at room temperature. The cutting solution and aCSF were continuously supplied with 95% O_2_–5% CO_2_.

### 4.3. Electrophysiology

Whole-cell patch-clamp recordings were carried out in a submerged recording chamber on the stage of differential interference contrast video microscope (BX-51WI; Olympus, Tokyo). The brain slice including the agranular insular cortex was placed on the recording chamber, and was perfused with aCSF at a solution flow rate of 2 mL/min. The recordings were conducted under visual guidance and carried out from visually identified PNs in layer V of the agranular insular cortex with a MultiClamp 700B amplifier (Molecular Devices, Foster City, CA, USA). The recordings were performed at temperature of 30–32 °C. The glass pipettes (4–6 MΩ) were filled with physiological intracellular recording solution containing (in mM) 132.5K-gluconate, 8.5 KCl, 14 Na-gluconate, 2 ATP-Mg, 0.3 GTP-Na_3_, 0.2 ethylene glycol tetraacetic acid (EGTA) and 10 4-(2-hydroxyethyl)-1-piperazineethanesulfonic acid (HEPES), which was adjusted to pH 7.3 with KOH. The sealing resistance was usually greater than 10 GΩ. The liquid junction potential (−10 mV) was compensated after recordings. The data of current and voltage responses were low pass filtered at 2 kHz using a Bessel filter, digitized at a sampling rate of 10 kHz with Digidata1440A (Molecular Devices) and saved in a computer hard disk.

Neurons were clamped −70 mV, and spontaneous excitatory postsynaptic currents (sEPSCs) were recorded in the presence of picrotoxin (100 μM). Electrical stimuli were delivered using a monopolar tungsten electrode placed in layer V of the agranular insular cortex (100–200 μm from the axis of the apical dendrite of layer V PNs. EPSCs were induced by extracellular repetitive stimulations at 0.33 Hz. The duration of single stimulation was 100 μs, and stimulation intensity was set to evoke EPSCs with an amplitude of around 100 pA. To produce LTP, 80 presynaptic stimulations were applied at 2 Hz, during the stimulations (40 s) recorded neurons were clamped at +30 mV [19,21]. The intensity and duration of stimulations were the same as those for control. The LTP-inducing protocol was applied within 12 min after obtaining the whole-cell configuration to prevent washout of the intracellular macromolecules that are crucial for the induction of synaptic plasticity [19,21]. LTP was induced in the presence of 100 μM picrotoxin in order to block GABA_A_ receptor-mediated inhibitory synaptic currents. Access resistance (15–20 MΩ) was continuously monitored during recordings.

To test the effects of adolescent nicotine exposure on the spike ability of a neuron, the relationship between current strength and spike number and inter-spike intervals of spikes were examined. Action potentials were induced by 1000 ms current injections that were initially applied at 30 pA and increased every 30 pA until 240 pA. Inter-spike intervals were analyzed for the spike trains evoked at the intensities of 180, 210 and 240 pA. The first inter-spike interval refers to the inter-spike interval between the first two spikes, and the steady-state inter-spike interval refers to the average inter-spike interval in the steady-state firings during the last 500 ms of the current pulse. The first and steady-state inter-spike intervals were calculated by averaging the respective inter-spike intervals obtained at three intensities. In addition, standard deviation of spike timing (SDST) which reflects precision of spike timing was analyzed for the spike trains evoked at the intensity of 180 pA. The SDST was calculated by using 20 recordings of spike trains evoked by repetitive stimulations. The first to tenth SDST were compared between saline-treated and nicotine-treated mice. Furthermore, the spike current threshold for initiating a spike and refractory period were also investigated. In this study, the spike current threshold refers to the current threshold, and a spike was evoked by 3 ms current injections. The spike current threshold was determined by increasing current stimulation intensity until evoking a spike at 50% probability. The refractory period is measured by applying 3 ms paired current pulses. The current stimulation intensity was 1.2 times larger than the spike current threshold. By changing the interval of paired current pulses, refractory period was defined as the time from a complete spike to the next one [24].

### 4.4. Drug Application

The NMDA receptor antagonist (AP-5) and the non-NMDA receptor antagonist (DNQX) were bath-applied at 50 μM and 10 μM, respectively. The GABA_A_ receptor antagonist (picrotoxin) was bath-applied at 100 μM. These chemicals were obtained by Sigma-Aldrich (St. Louis, MO, USA).

### 4.5. Data Analysis

Data analysis was carried out similar to our previous study [19]. Analysis of sEPSCs (frequency and amplitude) was performed using Minianalysis software (Synaptosoft, Decatur, GA, USA). The data were presented as the mean ± S.E. LTP was evaluated as the percentage increase in EPSC amplitude when mean EPSC amplitudes obtained at 25–30 min after LTP-inducing stimulation and at 5–10 min of control responses were compared. Statistical significance was accessed by paired and unpaired Student’s *t*-test and two-way repeated-measures ANOVA followed by Fisher’s protected least significant difference (LSD) post-hoc test. Student’s *t*-test was used when the data showed normal distribution. Statistical results were displayed exact *p* values, except when *p* was < 0.001. When a *p*-value was less than 0.05, results were considered statistically significant.

## Figures and Tables

**Figure 1 ijms-23-00034-f001:**
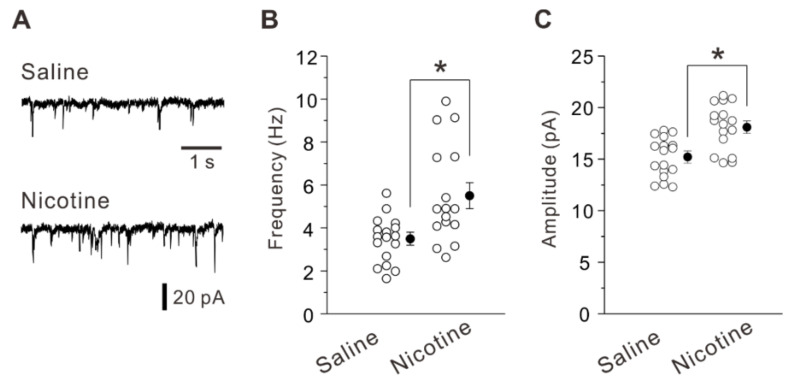
Enhanced excitatory synaptic transmission caused by adolescent nicotine exposure. (**A**) Sample traces of sEPSCs obtained from layer V PNs of a saline-treated mouse (top) and a nicotine-treated mouse (bottom). (**B**,**C**) Summary data of frequency (**B**) and amplitude (**C**) of sEPSCs obtained from PNs of saline-treated mice (n = 17 neurons/6 mice) and nicotine-treated mice (n = 16 neurons/7 mice). Unpaired *t*-test, *: *p* < 0.003.

**Figure 2 ijms-23-00034-f002:**
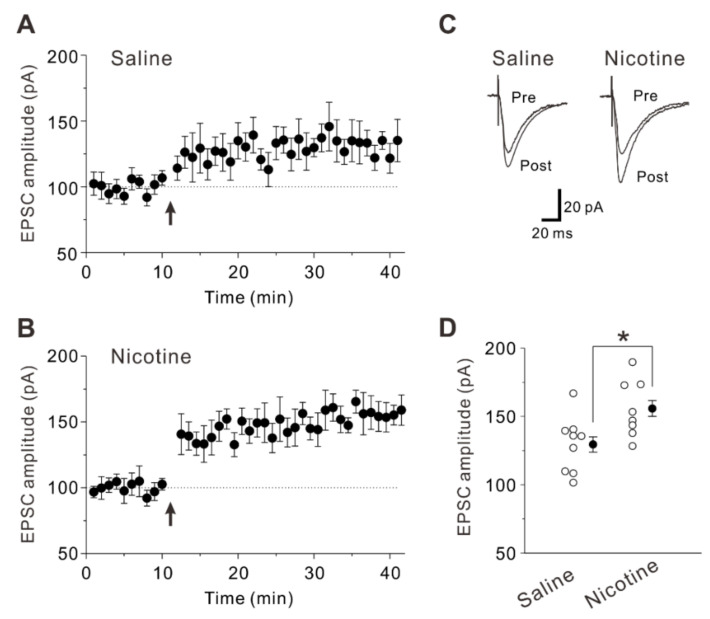
Enhanced synaptic potentiation caused by adolescent nicotine exposure. (**A**) Synaptic potentiation was induced in PNs of saline-treated mice (n = 9 neurons/6 mice) by LTP-inducing protocol (arrow: 80 presynaptic stimulations at 2 Hz with postsynaptic depolarization by clamping at +30 mV). (**B**) Synaptic potentiation was induced in PNs of nicotine-treated mice (n = 8 neurons/7 mice) by LTP-inducing protocol (arrow). (**C**) Sample traces of evoked EPSCs obtained by averages of ten current traces under control condition (pre) and 25–30 min after LTP-inducing protocol (post). (**D**) Summary data of normalized EPSC amplitude obtained in (**A**,**B**). Unpaired *t*-test, *: *p* = 0.019.

**Figure 3 ijms-23-00034-f003:**
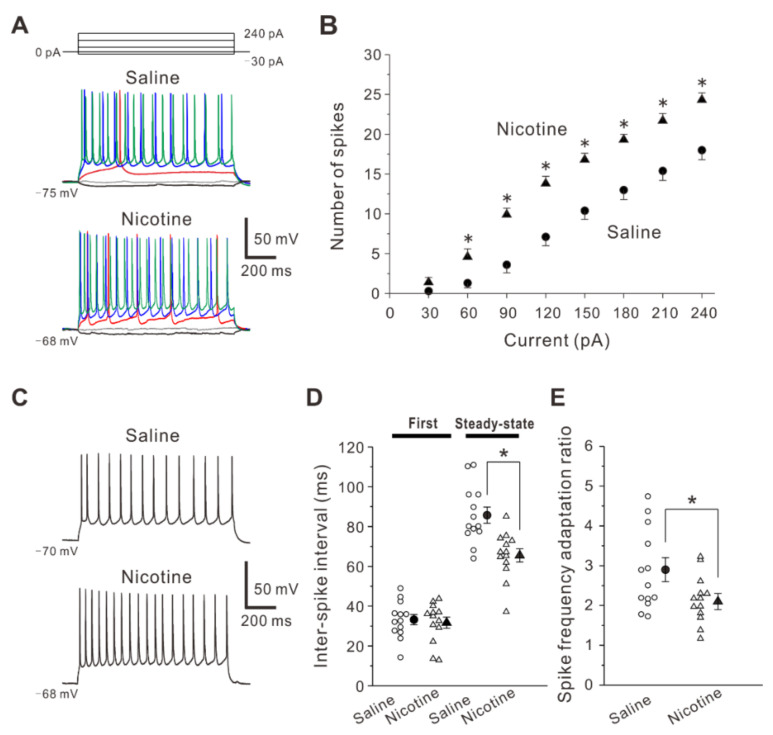
Effects of adolescent nicotine exposure on spiking ability and inter-spike interval. (**A**) Sample traces of voltage responses induced by current pulses obtained from PNs of a saline-treated mouse (top) and a nicotine-treated mouse (bottom). Black, gray, red, blue and green traces represent voltage responses induced by current pulses at the intensities of −30, 0, 60, 150 and 240 pA, respectively. (**B**) Relationship between current strength and spike number obtained from PNs of saline-treated mice (n = 16 neurons/6 mice) and nicotine-treated mice (n = 16 neurons/7 mice). Two-way repeated measures ANOVA post-hoc LSD, *: *p* < 0.05 between the two groups. (**C**) Sample traces of voltage responses induced by depolarizing current pulses at the intensity of 180 pA obtained from PNs of a saline-treated mouse (top) and a nicotine-treated mouse (bottom). (**D**) Summary data of first and steady-state inter-spike interval obtained from PNs of saline-treated mice (n = 13 neurons/6 mice) and nicotine-treated mice (n = 13 neurons/7 mice). Unpaired *t*-test, *: *p* < 0.001. (**E**) Summary data of spike frequency adaptation ratio obtained from PNs of saline-treated mice (n = 13 neurons/6 mice) and nicotine-treated mice (n = 13 neurons/7 mice). Unpaired *t*-test, *: *p* = 0.034.

**Figure 4 ijms-23-00034-f004:**
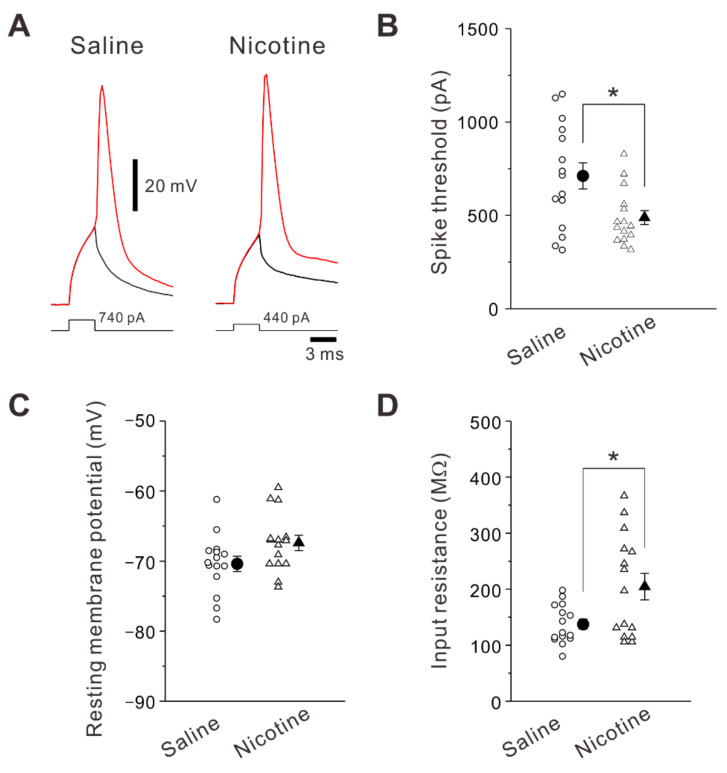
Effects of adolescent nicotine exposure on intrinsic membrane properties. (**A**) Sample traces of voltage responses to depolarizing current pulses at spike current threshold that induce spikes at 50% probability in PNs of a saline-treated mouse (left) and a nicotine-treated mouse (right). (**B**–**D**) Summary data of spike current threshold (**B**), resting membrane potential (**C**), and input resistance (**D**) obtained from PNs of saline-treated mice (n = 15 neurons/6 mice) and nicotine-treated mice (n = 15 neurons/7 mice). Unpaired *t*-test, *: *p* < 0.02.

**Figure 5 ijms-23-00034-f005:**
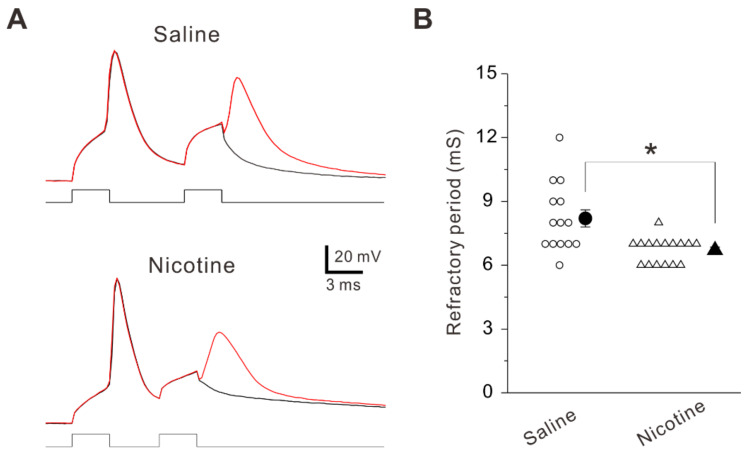
Effects of adolescent nicotine exposure on spike refractory period. (**A**) Sample traces of voltage responses to two successive current pulses obtained from PNs of a saline-treated mouse (top) and a nicotine-treated mouse (bottom). (**B**) Summary data of refractory period obtained from PNs of saline-treated mice (n = 14 neurons/6 mice) and nicotine-treated mice (n = 16 neurons/7 mice). Unpaired *t*-test, *: *p* = 0.002.

**Figure 6 ijms-23-00034-f006:**
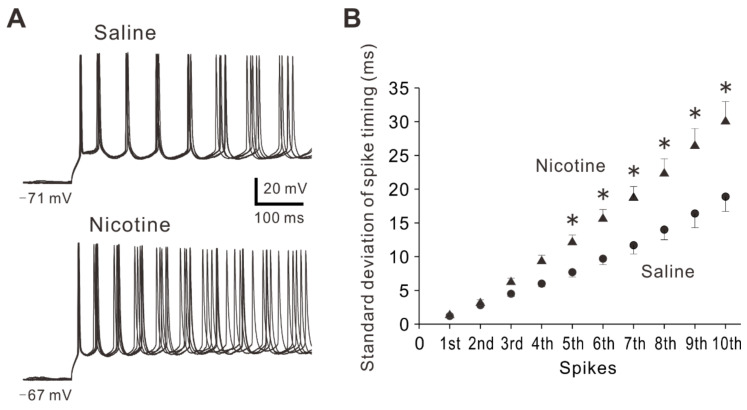
Effects of adolescent nicotine exposure on SDST. (**A**) Sample traces of voltage responses evoked by depolarizing current pulses at the intensity of 180 pA obtained from PNs of a saline-treated mouse (top) and a nicotine-treated mouse (bottom). (**B**) Relationship between spike number and SDST obtained from PNs of saline-treated mice (n = 16 neurons/6 mice) and nicotine-treated mice (n = 15 neurons/7 mice). Two-way repeated measures ANOVA post-hoc LSD, *: *p* < 0.05.

**Figure 7 ijms-23-00034-f007:**
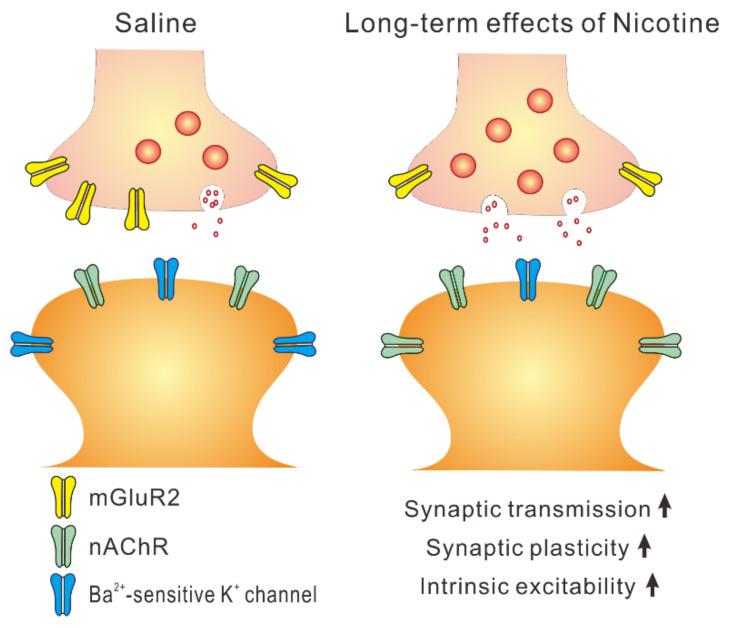
A possible model for long-term effects of adolescent nicotine exposure in layer V PNs of the mice insular cortex.

## Data Availability

The data presented in this study are available in request from the corresponding author.

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
