# Peer review of "Nicotine Exposure during Adolescence Leads to Changes of Synaptic Plasticity and Intrinsic Excitability of Mice Insular Pyramidal Cells at Later Life"

_ijms, 2021, doi:10.3390/ijms23010034_

Round 1

Reviewer 1 Report

There needs to be some clarification of the nicotine treatment used and that it most likely represents a pulsatile high dose twice a day. The papers referenced for dosage, Epping-Jordan  et. al. and Penton et al. used a dosage similar to that used here. However, those papers used osmotic mini-pumps and in rats. The half-life of nicotine in rats is about an hour while in mice it is about 10 minutes.  So the steady state levels they claim match that of human smokers wouldn't  be obtained using that treatment in mice. This treatment however may represent more accurately smoking patterns in adolescent humans. How nicotine may produce these effects was discussed, but a figure or model summarizing  the possible effects, including possible long term changes in gene or surface nAChR expression may be useful. 

Overall the results seem sound and intriguing. It is not clear how many cells were recorded from, the number of mice is mentioned but not how many recordings from each (or did  the n  value represent  the  number of neurons recorded from nicotine treated or control mice , not the number of mice). Please clarify.

Reviewer 2 Report

  1. The manuscript presents interesting data on the action of nicotine administered in vivo and analyzed ex vivo in layer V pyramidal neurons. Synaptic plasticity and neuronal  excitability are analyzed with sound electrophysiological techniques. Some points of this manuscript need revision and expansion, as presented in the detailed comments here below.
  2. The title should indicate that the work has been carried out in mice.
  3. The abstract should indicate the animal species used
  4. One of the problems of this article is that human and animals findings reported in Introduction and Discussion sections are not specified, and it is not clear if comparison and integration of human and animals results may be valid. New findings with imaging and specific probes indicate that there is not always concordance between animal and human findings in the central nervous system.
  5. Add PNs, to the abbreviations list since it is used many times and is not included.
  6. Line 117, is it “spiking”?
  7. Line 147, indicate the meaning of the red tracings shown in Fig. 3 A in saline-treated mouse (top) and a nicotine-treated mouse (bottom). Somewhere it should be indicated if the superimposed tracings correspond to the all the current stimulations reported.
  8. Line 205, it should be “and « SD » of spike timing (otherwise redundant)
  9. Lines 241-242, it should be indicated “in rodents”
  10. Line 290 “and these changes well reflects” it should be “and these changes reflect well “
  11. Line 328, indicate the reason why the study was performed only on male mouse.
  12. Line 329, indicate the weight of male mice used before and along the days of treatment (add a graph)
  13. Line 331, indicate the nicotine salt used
  14. The dose of nicotine used is extremely elevated for a mouse of 20 g weight, and has been used for studies in rats, justify the use of such elevated dose, and how the dose was chosen, other concentrations were tested?
  15. Line 353 it should be “microscope” since it refers to the “microscope stage”
